# Evaluation of NEWS2 response thresholds in a retrospective observational study from a UK acute hospital

Tanya Pankhurst,[1] Elizabeth Sapey ,[2,3] Helen Gyves,[4] Felicity Evison ,[4] Suzy Gallier,[4,5] George Gkoutos,[6] Simon Ball [7,8]

¹Renal Medicine, University Hospitals Birmingham NHS Foundation Trust, Birmingham, UK
²Institute of Inflammation and Ageing, University of Birmingham, Birmingham, UK
³PIONEER Hub, University of Birmingham, Birmingham, UK
⁴Health Informatics, University Hospitals Birmingham NHS Foundation Trust, Birmingham, UK
⁵PIONEER Technical Director, University of Birmingham, Birmingham, UK
⁶University of Birmingham, Birmingham, UK
⁷Better Care, Health Data Research, London, UK
⁸Chief Medical Officer, University Hospitals Birmingham NHS Founation Trust, Birmingham, UK

**Correspondence to**
Dr Elizabeth Sapey;
e.sapey@bham.ac.uk

## ABSTRACT

**Objective** Use of National Early Warning Score 2 (NEWS2) has been mandated in adults admitted to acute hospitals in England. Urgent clinical review is recommended at NEWS2 ≥5. This policy is recognised as requiring ongoing evaluation. We assessed NEWS2 acquisition, alerting at key thresholds and patient outcomes, to understand how response recommendations would affect clinical resource allocation.

**Setting** Adult acute hospital in England.

**Design** Retrospective observational cohort study.

**Participants** 100 362 consecutive admissions between November 2018 and July 2019.

**Outcome** Death or admission to intensive care unit within 24 hours of a score.

**Methods** NEWS2 were assembled as single scores from consecutive 24-hour time frames, (the first NEWS2 termed 'Index-NEWS2'), or as all scores from the admission (termed All-NEWS2). Scores were excluded when a patient was in intensive care, in the presence of a decision not to attempt cardiopulmonary resuscitation, or on day 1 of elective admission.

**Results** A mean of 4.5 NEWS2 were acquired per patient per day. The outcome rate following an Index-NEWS2 was 0.22/100 patient-days. The sensitivity of outcome prediction at Index-NEWS2 ≥5=0.46, and number needed to evaluate (NNE)=52. At this threshold, a mean of 37.6 alerts/100 patient-days would be generated, occurring in 12.3% of patients on any single day. Threshold changes to increase sensitivity by 0.1, would result in a twofold increase in alert rate and 1.5-fold increase in NNE. Overall, NEWS2 classification performance was significantly worse on Index-scores than All-scores (c-statistic=0.78 vs 0.85; p<0.001).

**Conclusions** The combination of low event-rate, high alert-rate and low sensitivity, in patients for cardiopulmonary resuscitation, means that at current NEWS2 thresholds, resource demand would be sufficient to meaningfully compete with other pathways to clinical evaluation. In analyses that epitomise in-patient screening, NEWS2 performance suggests a need for re-evaluation of current response recommendations in this population.

## INTRODUCTION

The use of early warning scores (EWS's) has been widely advocated, to integrate physiological parameters into a single actionable

---

## Strengths and limitations of this study

► All admissions to an acute hospital within the study time frame are included, providing the basis for a detailed understanding of the consequences of National Early Warning Score 2 (NEWS2)-based policy.

► A precise definition of decision not to attempt cardiopulmonary resuscitation decisions over the course of an admission informed our analysis, in order to maintain correspondence with decision-making and treatment options available clinically.

► The evaluation of NEWS2 as a single score from discrete 24-hour time frames, provides an assessment of classification performance which epitomises its role in screening patients between routine reviews, moderating the effects of including large numbers of scores acquired from those already identified to be at clinical risk.

► The analysis is limited by the fact that this is a single-centre study, however, the underlying data are consistent with other reports from acute hospitals in the National Health Service.

► The analysis excludes day 1 of elective admission from outcome-based analysis, because deranged physiology was predicted to occur following planned intervention on that day.

---

output. In 2017, the Royal College of Physicians (RCP) published a modified National Early Warning Score (NEWS), referred to as NEWS2.[1] NEWS2 is a scoring system based on six physiological parameters. It is associated with specific clinical response recommendations, including urgent clinical review at a key threshold NEWS2 score ≥5. This requires attendance by clinicians with competence in the assessment and treatment of acutely ill patients, and where necessary, escalation to a team with critical care competencies.[1] At NEWS2 score ≥7, this is uplifted to assessment by a team with those critical care competencies. In 2019, the National Health Service in England (NHSE) mandated the use of NEWS2 for all adults in acute hospitals and

widened its application to include screening for sepsis.[2] This extends its use significantly beyond the evaluation of acute admissions, for which NEWS was initially developed and validated.[3–8] The National Institute for Health and Care Excellence (NICE) has identified the need for further evaluation of NEWS2, to ensure no adverse consequences from its roll out across the National Health Service (NHS).[9] This is in the context of limited evidence of survival benefit from an EWS triggered response,[8 10–16] or definition of the resource required to deliver a response.[17] Since electronic systems directly link a threshold EWS to a response recommendation,[18] high rates of alerting may arise, unmoderated by human intervention,[19 20] creating an opportunity for clinical resource to be diverted in ways that could be counterproductive.[9]

Given a requirement to significantly alter clinical practice,[1 2] we evaluated the performance of NEWS2, across the in-patient population of an acute hospital. This included a description of the rate of NEWS2 data acquisition, the rates at which key threshold scores were met, and their relationship to outcome. This was with a view to understanding how NEWS2-based recommendations could affect the distribution of resource when put into practice, taking into account factors which may modify the outcome, such as resuscitation status, or those which may amplify demand, such as recurrent alerting.[9]

## METHODS
### Setting
The Queen Elizabeth Hospital Birmingham (QEHB) is an NHS, urban, adult, acute hospital in England with 1269 beds including 80 level 2/3 intensive care unit (ICU) beds, an emergency department that assesses >300 patients per day, and a mixed secondary and tertiary practice that includes all major adult specialities with the exception of obstetrics and gynaecology. The Electronic Health Record (EHR) at QEHB (PICS, Birmingham Systems) contains time-stamped, structured records that include demography, location, time of admission and discharge, physiological measurements supporting NEWS2 and Standard Early Warning Score (SEWS) (see online supplemental table S1) and do-not-attempt-cardiopulmonary-resuscitation (DNACPR) decisions subject to regular review underpinned by the clinical decision support system. NEWS2 observations were recorded by trained healthcare staff with devices used and maintained in accordance with the Medicines and Healthcare Products Regulatory Agency guidance.[21]

NEWS2 data collection was electronically mandated, while alerting continued to use established SEWS thresholds,[22] (coincidentally facilitating an assessment of NEWS2 performance at the key action threshold ≥5, minimally disrupted by triggered clinical responses (see online supplemental tables S2 and S3). Existing standards included a maximum interval between EWS acquisition=12 hours, progressive alerting to ward-based staff and automated escalation to a 24/7 critical care outreach team at threshold SEWS.

### Cohort and definitions
All hospital spells (continuous stay in a hospital bed), between 00.00 on 1 November 2018 and 23.59 on 31 July 2019, were evaluated to discharge (99.6%), or to 56 days postadmission (0.4%) if that was earlier. Initial emergency department assessments, prior to admission, were not included. An adapted consort diagram is shown in figure 1. Admissions were identified as emergency or elective from the mandatory provider spell admission method code. A composite outcome event was defined as the first of unplanned admission to ICU (type 1 and 2 of the NHS critical care minimum dataset[23]) or death of the patient, within 24 hours of an NEWS2 score.

In the primary analysis, each spell was divided into consecutive 24-hour post-admission days starting at the time of admission. As shown in figure 1, for admission time T, consecutive 24-hour periods end at T + (n x 24 hours); where n is the nominal postadmission day. The first NEWS2 recorded in each post-admission day, at time $t_n$ was termed the Index-NEWS2 score. An overlapping, patient-day variable was defined between $t_n$ and $t_n$ +24 hours. The first outcome occurring in the patient-day was linked to the preceding Index-NEWS2. This design was used to ensure a single NEWS2 score was captured from discrete, consecutive, 24-hour time periods and that the outcome was assessed over 24 hours.

In addition to Index-NEWS2, analyses were undertaken using all the scores recorded in a given time period (termed All-NEWS2)[24] and using the highest score on each postadmission day (termed Highest-NEWS2). NEWS2 scores were not eligible for inclusion if at the time the score was acquired, the patient was in ICU or had a DNACPR in place. This was to achieve internal consistency when using admission to ICU as part of the composite outcome, since decisions not to resuscitate are highly concordant with ceiling of care not including ICU. On return from ICU or if a DNACPR was revoked, subsequent scores were included in the analysis. No score was calculated when patients underwent an operative procedure away from the ward.

Day 1 postelective admission was excluded from outcome-based analysis because it was known that almost all were admitted for surgery later that day, after an NEWS2 was recorded. Any relationship with outcome might then be confounded by decisions not to proceed to intervention on that day, informed by NEWS2 or its component observations. All other patient-days were eligible for analysis.

Patient demographics are reported for the first admission in the study period. Additional context is provided by reporting bed occupancy at midday across the study period (see online supplemental table S4).

### Analysis of EWS performance
To evaluate the performance of different NEWS2 threshold scores, a patient with an NEWS2 ≥threshold score was

**A**

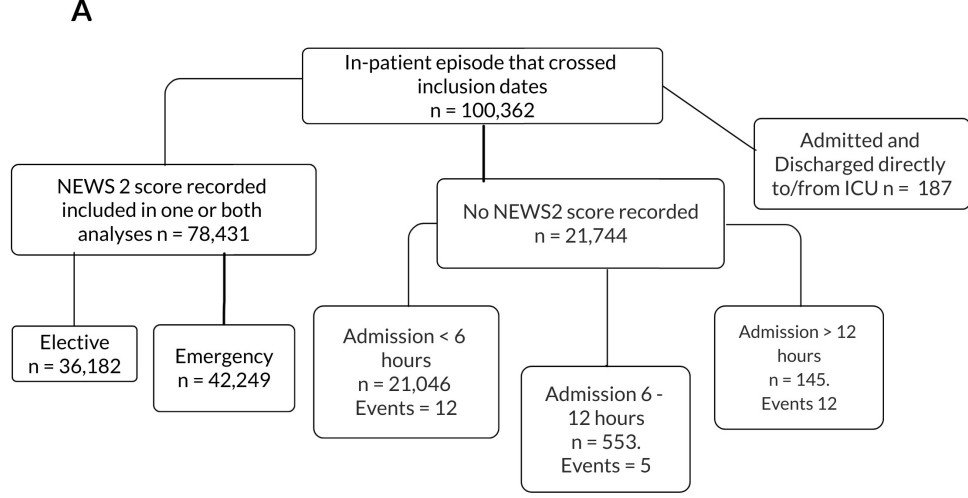

**B**

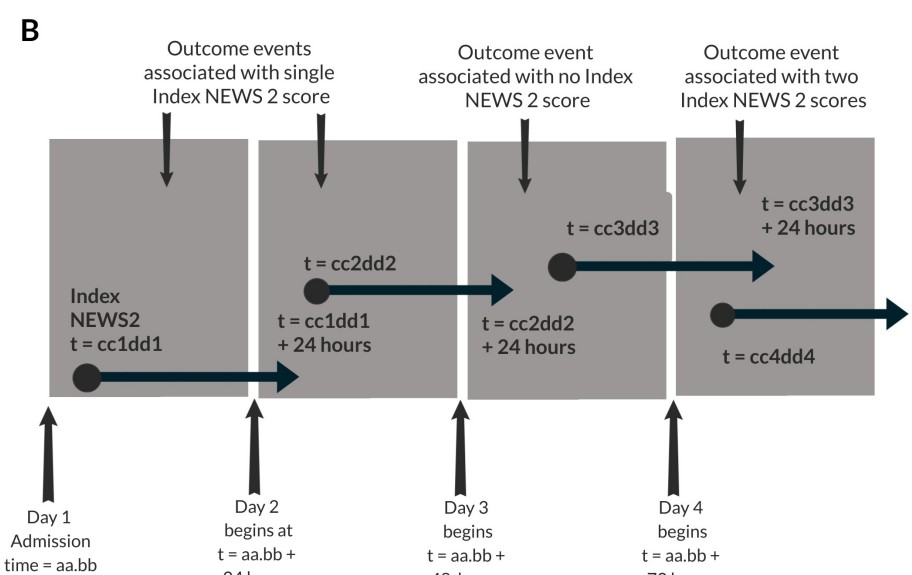

**Figure 1** Modified consort diagram and overview of study design. (A) Of 100 762 consecutive in-patient spells, 78 431 (77.8%) recorded at least one NEWS2 score whilst admitted outside of ICU. Of the remaining 21 931, 0.8% were admitted and discharged from ICU and so were never eligible for inclusion in the analysis. Of the others in which no NEWS2 score was recorded 96.8% had a length of stay <6 hours, 99.3% <12 hours (and 0.3% ≥24 hours). There were 29 deaths or admission to ICU associated with spells in which no NEWS2 was recorded. (B) Postadmission days were defined as consecutive 24-hour periods beginning at the time of admission T. The Index-NEWS2 score was the first recorded in each of these periods at time t (black circle). A distinct overlapping 'patient-day' variable was then defined for each NEWS2 score between t and t +24 hours during which, occurrence of a composite outcome event (black diamond) was recorded and linked to the preceeding Index-NEWS2. Since the exact time t, that the Index-NEWS2 was recorded varies on each day, a small number of outcome events may be associated with no Index-NEWS2 score (3.5%) as illustrated during day 3, or with two Index-NEWS2 scores (9.6%) as illustrated during day 4. ICU, intensive care uit; uNEWS2, National Early Warning Score 2.

defined as predicted positive (P) and a patient with an NEWS2 <threshold score as predicted negative (N). At different NEWS2 threshold values, performance metrics were calculated from four different groups: true positive, when an outcome event was correctly predicted, false positive when an outcome event was predicted but did not occur; with true negative and false negative following suit. Receiver operating characteristics (ROCs) were derived from these metrics, for Index-NEWS2 and All-NEWS2.

A range of performance metrics were calculated including two identified to be particularly suited to the representation of EWS performance, namely the number needed to evaluate (NNE) and the Alert Rate.[25] NNE is the number of patients meeting a threshold NEWS2 score, to include one who then sustains an outcome event, defined as follows:

$$NNE = \frac{TP+FP}{TP} = \frac{1}{PPV}$$

The alert rate is the number of threshold scores (alerts that would be generated) per 100 inpatients per day, defined as follows:

$$Alert\ Rate = \frac{TP+FP}{TP+FP+TN+FN}\ x\ number\ of$$
$$NEWS2\ scores\ acquired\ (/100\ patients/day)$$

The NNE and alert rate were plotted against the sensitivity as previously discussed.[25] For the Index-NEWS2, an outcome event rate per 100 patient-days, was calculated (overall event rate) as follows:

$$Overall\ Outcome\ Event\ Rate =$$
$$\frac{TP+FN}{TP+FP+TN+FN}\ x\ 100\ /100\ patient\ days$$

For the Index-NEWS2, an outcome event rate at any given threshold score (alerted event rate) per 100 patient-days, was calculated as follows:

$$Alerted\ Outcome\ Event\ Rate =$$
$$\frac{TP}{TP+FP+TN+FN}\ x\ 100\ /100\ patient\ days$$

For the purpose of reporting any daily rate, time of discharge was not considered, a whole day was counted.

### Estimation of clinician resource requirement
The clinical resource required to support response recommendations was illustrated by assuming 1 hour of healthcare professional time per clinical evaluation, informed by a report of a rapid clinical response team on a surgical ward in the Netherlands.[26] The number of healthcare professionals required/100 beds occupied was calculated based on a 40-hour working week and 18% overhead for leave (=1760 hours/year).

### Statistical analysis
All statistical analyses were undertaken in STATA SE V.15.1. Normally distributed variables are represented as mean±SD others as median and IQR. Bootstrap analyses of the ROC were undertaken with 10 000 repetitions of 10 000 patient-days. There were no adjustments for multiple comparisons and all p values are reported.

## RESULTS
### Patients and admissions
There were 100 362 admissions across 273 days. In 21 744, no NEWS2 was recorded, 21 599 (99.3%) of which were associated with a short length of stay <12 hours (figure 1A). The other 78 431 admissions in which NEWS2 was recorded, occurred in 52 214 patients (table 1).

### DNACPR and outcome
A DNACPR decision was made in 4621 (4.6%) admissions. This resulted in a DNACPR that was active in 170±15 (13%) in-patients at midday (online supplemental table S4) of the online supplement). Of 1076 deaths not on ICU, 943 (87.6%) occurred in those with a DNACPR decision, in 834 of whom this had been in place >24 hours.

### Index-NEWS2 recording and outcome
A DNACPR was active throughout the course of 367 admissions. In the remaining 77 877 admissions, 294 602 Index-NEWS2 were recorded. They were associated with 715 outcome events in the following 24 hours (154 deaths and 561 admissions to ICU). Of 10 731 post-admission days in which no NEWS2 was calculated, 5786 (53.9%) involved discharge later that day, while 4424 (41.2%) comprised a whole day in which the patient was eligible for inclusion and in which no outcome event occurred (see online supplemental figure S1).

### Day 1 postelective admission
Exclusion of day 1 of elective admission from the analysis of Index-NEWS2 associated outcome, was supported by finding that 94.7% of admissions were followed by a planned procedure, that this occurred within 12 hours, and all outcome events followed a procedure. (The performance of NEWS2 across day 1 post elective admission is shown in online supplemental figure S2A).

### Classification performance of NEWS2
Across all other days, 1 162 824 All-NEWS2 scores were recorded on 258 678 postadmission days, so a mean of 4.5 NEWS2 were recorded per day. A total of 580/258 678 (0.22%) Index-NEWS2 were associated with an outcome event. A total of 5284/1 162 824 (0.45%) All-NEWS2 were associated with an outcome event, since a mean of 9 NEWS2 were recorded in the 24 hours prior to an outcome event. The c-statistic of outcome prediction was higher when derived from All-NEWS2 compared with Index-NEWS2 (0.85 vs 0.78; p<0.001; see online supplemental figure S2B).

### Alert rate and NNE associated with Index-NEWS2
Table 2 presents Index-NEWS2 performance. The Index-NEWS2 Alert Rate is shown in table 3. Figure 2A plots this Alert Rate against sensitivity for the Index-NEWS2 score. Around current key clinical response thresholds, the relationship was log-linear: a 2.0x increase in the Alert Rate for an increase in sensitivity of 0.1 (Alert Rate = $0.19e^{7.18 x sensitivity}$ between NEWS2 ≥10 and ≥2). At the key action threshold NEWS2 ≥5 the Alert Rate generated by an Index-NEWS2=5.3/100 patients/day, at which score the Index-NEWS2 sensitivity=0.46.

Figure 2 shows the relationship between the NNE and sensitivity. Around current key clinical response thresholds, the relationship was log-linear: a 1.5× increase in NNE to increase sensitivity by 0.1 (NNE=8.7 $e^{4.14 x sensitivity}$ between NEWS2 ≥10 and ≥2). At the key action threshold NEWS2 score ≥5, the NNE=52.

### Alert rate associated with All-NEWS2
The Alert Rate at a threshold Index-NEWS2 score represents the rate derived from that single score. However, a mean of 4.5 NEWS2 scores were recorded per postadmission day. The alert rate derived from All-NEWS2 scores is shown in table 3. Figure 2A plots the alert rate generated by All-NEWS2 against the sensitivity

| Table 1 Demographic and clinical characteristics of patients | | |
| --- | --- | --- |
| | **Elective** | **Emergency** |
| No of patients with one or more included admission eligible for analysis (first admission) | 22 538 | 29 122 |
| Mean age/years (at time of first admission) | 53.9±18.1 | 56.2±21.7 |
| No of male patients (percentage) | 12 142 (53.9%) | 13 800 (47.4%) |
| Ethnicity | | |
| White | 16 153 (71.7%) | 19 596 (67.3%) |
| South Asian | 2642 (11.7%) | 4203 (14.4%) |
| Black | 943 (4.2%) | 1489 (5.1%) |
| Other | 806 (3.6%) | 1434 (4.9%) |
| Not known | 1994 (8.8%) | 2400 (8.2%) |
| Admitting specialty on first admission | | |
| General medicine | 1545 (6.9%) | 20 174 (69.3%) |
| General surgery | 2150 (9.5%) | 3061 (10.5%) |
| Trauma and orthopaedics | 2307 (10.2%) | 1486 (5.1%) |
| Neurosurgery | 1866 (8.3%) | 639 (2.2%) |
| Urology | 1411 (6.3%) | 587 (2.0%) |
| Cardiology | 1821 (8.1%) | 469 (1.6%) |
| Clinical and medical oncology | 1263 (5.6%) | 549 (1.9%) |
| Ear, nose and throat | 1401 (6.2%) | 389 (1.3%) |
| Plastic surgery | 2138 (9.5%) | 304 (1.0%) |
| Maxillo-facial surgery | 849 (3.8%) | 252 (0.9%) |
| All others | 5787 (25.7%) | 1212 (4.2%) |
| Mean no of admissions/patient | 1.6 | 1.5 |
| No of admissions in which the patient was ineligible for analysis throughout an admission | 60 | 494 |

The demographics, mode of admission and admitting specialty on first admission, of the 52 214 patients that were subject to NEWS2 analysis, contributing 36 182 elective and 42 249 emergency admissions. In 554 admissions the patient was ineligible for inclusion in analysis throughout the spell (187 due to admission and discharge from ICU and 367 due to a DNACPR decision).
DNACPR, do-not-attempt-cardiopulmonary-resuscitation; ICU, intensive care unit; NEWS2, National Early Warning Score 2.

of Index-NEWS2, across different thresholds. Around current clinical response thresholds, the relationship was log-linear: a 1.8× increase in the Alert Rate for an increase in sensitivity of 0.1 (alert rate = $2.12e^{6.14 \times sensitivity}$ between NEWS2 ≥10 and ≥2). At the key action threshold NEWS2 score ≥5, the alert rate generated by All-NEWS2=37.6/100 patients/day.

### Alert rate associated with highest-NEWS2
The alert rate generated by the Highest NEWS 2 score in each postadmission day is also shown in table 3. Figure 2A plots the alert rate generated by the Highest-NEWS2, against the sensitivity of the Index-NEWS2, across different thresholds. Around the current clinical response thresholds, the relationship was log-linear: a 1.8× increase in the alert rate for an increase in sensitivity of 0.1 (Alert Rate = $0.73e^{6.07 \times sensitivity}$ between NEWS2 ≥10 and ≥2). At the key action threshold NEWS2 score ≥5, the alert rate generated by the Highest-NEWS2=12.3/100 patients/day.

### Incremental clinical resource required by changes to NEWS2 threshold
The number of clinicians required to support responses deployed at the different NEWS2 thresholds, is also estimated in table 3. This is derived from the alert rates defined by the Index-NEWS2, All-NEWS2 and the Highest-NEWS2 at each threshold, with 1 hour assigned per clinical response deployed. Thus, at the key action threshold NEWS2 score ≥5, demand for clinician resource was calculated to be respectively 1.1, 7.8 and 2.6 whole time equivalent (WTE) clinicians/100 in-patients.

### Calibration of NEWS2
Calibration was not assessed as NEWS2 does not estimate absolute risk.[27] Usefully, integer changes in NEWS2 threshold were associated with approximately equal relative changes in the OR of outcome events, across a wide range of scores (see online supplemental figure S3A and B).

**Table 2** Performance of Index-NEWS 2

| Threshold NEWS2 | No of Index-NEWS2 meeting threshold | No of composite outcome events associated with Index-NEWS2 at threshold | Alert rate/100 patients/postadmission day (95% CI) | Sensitivity (%) (95% CI) | Specificity (%) (95% CI) | PPV (%) (95% CI) | NPV (%) (95% CI) | Alerted outcome event rate/100 patient-days | NNE | LR+ | LR− |
|---|---|---|---|---|---|---|---|---|---|---|---|
| ≥1 | 184867 | 534 | 71.5 (71.1 to 71.8) | 92.1 (89.6 to 94.1) | 28.6 (28.4 to 28.8) | 0.3 (0.3 to 0.3) | 99.9 (99.9 to 100) | 0.21 (0.19–0.22) | 347 | 1.3 | 0.28 |
| ≥2 | 101364 | 453 | 39.2 (38.9 to 39.4) | 78.1 (74.5 to 81.4) | 60.9 (60.7 to 61.1) | 0.4 (0.4 to 0.5) | 99.9 (99.9 to 99.9) | 0.17 (0.16–0.19) | 224 | 2.0 | 0.36 |
| ≥3 | 53713 | 383 | 20.8 (20.6 to 20.9) | 66.0 (62.0 to 69.9) | 79.3 (79.2 to 79.5) | 0.7 (0.6 to 0.9) | 99.9 (99.9 to 99.9) | 0.15 (0.13–0.16) | 141 | 3.2 | 0.43 |
| ≥4 | 26484 | 318 | 10.2 (10.1 to 10.4) | 54.8 (50.7 to 58.9) | 89.9 (89.7 to 90.0) | 1.2 (1.1 to 1.3) | 99.9 (99.9 to 99.9) | 0.12 (0.11–0.14) | 84 | 5.4 | 0.50 |
| ≥5 + Single=3 | 21833 | 288 | 8.4 (8.3 to 8.6) | 49.7 (45.5 to 53.8) | 91.7 (91.5 to 91.8) | 1.3 (1.2 to 1.5) | 99.9 (99.9 to 99.9) | 0.11 (0.10–0.12) | 76 | 6.0 | 0.55 |
| ≥5 | 13793 | 268 | 5.3 (5.2 to 5.4) | 46.2 (42.1 to 50.4) | 94.8 (94.7 to 94.8) | 1.9 (1.7 to 2.2) | 99.9 (99.9 to 99.9) | 0.10 (0.09–0.12) | 52 | 8.8 | 0.57 |
| ≥6 | 7252 | 216 | 2.8 (2.7 to 2.9) | 37.2 (33.3 to 41.3) | 97.3 (97.2 to 97.3) | 3.0 (2.6 to 3.4) | 99.9 (99.9 to 99.9) | 0.08 (0.07–0.10) | 34 | 13.7 | 0.65 |
| ≥7 | 3824 | 153 | 1.5 (1.4 to 1.5) | 26.4 (22.8 to 30.2) | 98.6 (98.5 to 98.6) | 4.0 (3.4 to 4.7) | 99.8 (99.8 to 99.8) | 0.06 (0.05–0.07) | 25 | 18.5 | 0.75 |
| ≥8 | 2095 | 105 | 0.8 (0.8 to 0.8) | 18.1 (15.1 to 21.5) | 99.2 (99.2 to 99.3) | 5.0 (4.1 to 6.0) | 99.8 (99.8 to 99.8) | 0.04 (0.03–0.05) | 20 | 23.5 | 0.83 |
| ≥9 | 1158 | 75 | 0.4 (0.4 to 0.5) | 12.9 (10.3 to 15.9) | 99.6 (99.6 to 99.6) | 6.5 (5.1 to 8.1) | 99.8 (99.8 to 99.8) | 0.03 (0.02–0.04) | 16 | 30.8 | 0.87 |
| ≥10 | 613 | 51 | 0.2 (0.2 to 0.3) | 8.8 (6.6 to 11.5) | 99.8 (99.8 to 99.9) | 8.3 (6.3 to 10.8) | 99.8 (99.8 to 99.8) | 0.02 (0.01–0.03) | 12 | 40.4 | 0.91 |

The Index-NEWS2 score is the first recorded in each consecutive 24-hour period postadmission. A total of 258678 Index-NEWS2 score were analysed from day 1 to day 56 of admission, excluding the first day of elective admission. There were 580 outcome events.

The alerted outcome event rate per 100 patient days is the number of outcome events following an Index-NEWS2 meeting the threshold NEWS2 score. The overall outcome event rate=0.22 per 100 patient-days.

LR−, negative likelihood ratio; LR+, positive likelihood ratio; NEWS2, National Early Warning Score 2; NNE, number needed to evaluate; NPV, negative predictive value; PPV, positive predictive value.

## DISCUSSION

In 2019, NHSE required that NEWS2 be used to monitor all adults in acute hospitals,[2] extending its scope to include screening for sepsis. This was linked to clinical response recommendations defined by the RCP, including a key threshold for urgent clinical review at NEWS2 ≥5.[1] NICE cautioned that when used in this way, there was a risk that NEWS2 based recommendations could result in new demand, which paradoxically might adversely affect overall care delivery.[9] A need for ongoing evaluation was identified. Our analysis was designed to provide a description of how NEWS2 implementation could affect the disposition of clinical resource, across an acute hospital in-patient population, considering the effects of response modifiers such as resuscitation status as well as repeated measurement in those already identified for increased levels of care.

In contrast to other studies, NEWS2 recorded at the time of a DNACPR was excluded from our analysis. This was possible because time stamped records of both were available in the EHR. Previous reports have excluded patients on end-of-life care pathways, however this had to be inferred from an absence of observations in the 24 hours prior to death.[27–29] Precise delineation of these two groups shows that the majority of deaths occurred in patients with a DNACPR in place for more than 24 hours, whereas ICU admissions occurred in patients without a DNACPR. Although this may be unsurprising, it is important when considering the translation of NEWS2 into clinical practice. Compared with previous reports,[27–29] there was a low event rate in the large group of patients who were eligible for outcome-based analysis, because the majority of deaths occurred in patients with a DNACPR. A low event rate in this group is one reason why NEWS2 may not perform as well as expected when translated into clinical practice.[9 15] It is evident in metrics sensitive to the event rate, such as the NNE. This analysis does not imply that NEWS2 is not applicable to patients with a DNACPR, but it does reflect difficulty in interpreting the real-world consequences of recommended thresholds, if these populations are not analysed separately.

Index-NEWS2 was used to assemble single scores from discrete postadmission time frames, to limit over-representation of scores recorded for clinical indication rather than routine screening. Clinical practice guidelines, including those associated with NEWS2,[1] require increased monitoring of physiological parameters to track those already identified to be at risk of further deterioration. This requires a progressive representation of risk, a different task to screening a population using routine observations. Our findings are relevant to the latter. The better classification performance of NEWS2 on All-scores compared with Index-scores implies that NEWS2 discriminates better on days in which data acquisition is more frequent.[30] This conclusion is apparently different to the influential report of Jarvis *et al*, which found that the c-statistic of NEWS and other scoring systems are little affected when using all or single scores per admission.[24]

**Table 3** Alert rates and clinician resource required to respond at threshold NEWS2

| Threshold NEWS2 | Index-NEWS2 Alert rate/100 patients/ post-admission day | WTE clinician/100 occupied beds to respond to Index NEWS2 at threshold | All-NEWS2 Alert rate/100 patients/ postadmission day | WTE clinician /100 occupied beds to respond to all NEWS2 at threshold | Highest-NEWS2 Alert rate/100 patients/ postadmission day | WTE clinician/100 occupied beds to respond to highest NEWS2 at threshold |
|---|---|---|---|---|---|---|
| ≥1 | 71.5 | 14.8 | 335.4 | 69.6 | 91.4 | 19.0 |
| ≥2 | 39.2 | 8.1 | 202.3 | 41.9 | 64.5 | 13.4 |
| ≥3 | 20.8 | 4.3 | 117.6 | 24.4 | 39.6 | 8.2 |
| ≥4 | 10.4 | 2.2 | 65.3 | 13.6 | 22.1 | 4.6 |
| ≥5 or single=3 | 8.4 | 1.7 | 52.9 | 11.0 | 19.3 | 4.0 |
| ≥5 | 5.3 | 1.1 | 37.6 | 7.8 | 12.3 | 2.6 |
| ≥6 | 2.8 | 0.6 | 22.2 | 4.6 | 7.1 | 1.5 |
| ≥7 | 1.5 | 0.3 | 12.9 | 2.7 | 4.1 | 0.9 |
| ≥8 | 0.8 | 0.2 | 7.6 | 1.6 | 2.5 | 0.5 |
| ≥9 | 0.4 | 0.1 | 4.4 | 0.9 | 1.5 | 0.3 |
| ≥10 | 0.2 | <0.1 | 2.4 | 0.5 | 0.9 | 0.2 |

The alert rate is the number of NEWS2 scores that reach threshold per 100 patients per postadmission day. NEWS2 score were analysed from day 1 to day 56 of admission, excluding the first day of elective admission. Index-NEWS2 reports the first NEWS2 recorded in each post-admission day. All-NEWS2 includes all scores recorded in the admission spell. Highest-NEWS2 reports the highest score recorded in each post-admission day. The WTE clinician resource required to service clinical response recommendations at different thresholds, assumed 1 hour per deployed response and an annual clinician workload=1760 hours. NEWS2 were excluded from analysis in the presence of a DNACPR decision or when the patient was on ICU.

DNACPR, do-not-attempt-cardiopulmonary-resuscitation; ICU, intensive care unit; NEWS2, National Early Warning Score 2; WTE, whole time equivalent.

However, their random selection of single scores across an admission would be expected to mirror the distribution of all scores. In contrast, a methodology employing single scores from a time frame typical of that between routine reviews[31 32] is less influenced by superior performance on scores obtained during times of high frequency monitoring. Arguably, our analysis of Index-NEWS2 more closely reflects its use as a screening tool in the interval between ward-round based routine assessment, which is typically 24 hours. This difference in performance is another reason why in clinical practice, NEWS2 might not perform as well as expected from the literature.[15]

Even at the lower action threshold ≥5, Index-NEWS2 sensitivity was less than 0.5. Other routes to clinical evaluation, such as routine or symptom-based assessment are therefore also likely to play an important role in identifying deteriorating patients for assessment. The resource with which to deliver these assessments must then compete with that demanded by NEWS2 triggered responses. Diversion of resource is one mechanism by which NEWS2 associated response recommendations could adversely affect patient care, particularly if alert rates are high.[9]

In our centre, alert rates would be high at current NEWS2 thresholds. This is consistent with data from other NHS acute hospitals.[27 29] A meaningful diversion of clinical resource away from other routes to evaluation is therefore possible.[9] Equally, alert fatigue could arise, thereby degrading clinical decision making.[33 34] In response to this, some centres have developed local policies to manage repeated alerting in some situations.[35]

Given that responses to All-NEWS2 may be modified in various ways, including prospective decision-making or censoring by ward-staff at the point of care,[36] we also

defined an alert rate based on the Highest-NEWS2 score each day. This is operationally equivalent to allowing one triggered response per person per day. Although it might underestimate the optimal response rate, this provides additional context with which to understand the boundaries of reasonable clinical resource deployment.

An optimal threshold score would usually be defined by health economic analysis.[8 37] This is not possible because there is little evidence quantifying survival benefit from EWS triggered responses,[8 10–17] nor is there a consistent account of the cost of current clinical response recommendations. We assigned 1 hour of clinician time per response, to illustrate the potential resource implications of different thresholds. This was based on a single informative study[26] identified by NICE guidance on emergency and acute care.[17] It evaluated a team with critical care competencies, most like that indicated for NEWS2 ≥7. We chose not to vary this resource attribution further, as our aim was to provide perspective on scale rather than precise definition of cost.

Although a health economic analysis is not possible, description of the Alert Rate and Event Rate provides some insight into the resource consequences of different NEWS2 thresholds, across an acute hospital in-patient population. For example, reducing the NEWS2 threshold from ≥7 to ≥5 would increase the alert rate from 4.1 to 12.3/100 postadmission days, thereby increasing the modelled demand for healthcare professionals from 0.9 to 2.6 WTE/100 in-patients. This resource estimate is based on Highest-NEWS2, so may be an underestimate. The number of Index-NEWS2 triggered responses followed by an event would increase by 0.04/100 patient-days, but 0.12 events/100 patient-days would still not have been predicted. Even as an approximation, these results

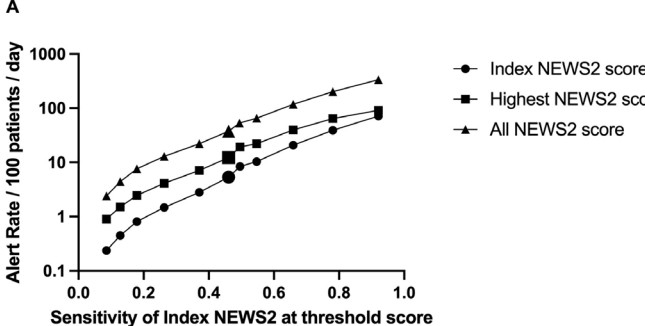

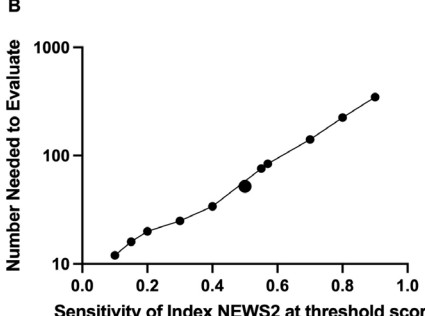

**Figure 2** Alert rate and number needed to evaluate for NEWS2. (A) Alert rate for Index, all and highest NEWS2 score versus sensitivity for Index-NEWS2 score The alert rate is the number of NEWS2 scores recorded at any given threshold, per 100 patients per postadmission day. NEWS2 scores from day 1 to 56 postadmission other than for day 1 of elective admission were evaluated. The larger marker denotes the key action threshold NEWS2 = or >5, to the right of which is the marker for NEWS2 = or >5 or single parameter = 3 (and then NEWS2 = or >4, = or >3, = or >2, = or >1), and to left of which is NEWS2 = or > 6 (and then NEWS2 = or >7, = or >8, = or >9, = or >10). (B) Number needed to evaluate versus sensitivity for Index-NEWS2 score. The number needed to evaluate is the number of patients to whom an Index-NEWS2 score based clinical response must be deployed at threshold, to include one who then sustains a linked composite outcome event. The composite outcome event was defined as the first of unplanned admission to ICU (type 1 and 2 of the NHS critical care minimum dataset) or death of the patient within 24 hours of an NEWS2 score. ICU, intensive care unit; NEWS2, National Early Warning Score 2; NHS, National Health Service.

reveal that small changes in current thresholds can result in demand for clinical resource that could meaningfully impact delivery through other care pathways.

Our analysis of NEWS2 excluded patients when there was a DNACPR decision. This is a diverse group of patients, including those approaching the end of life as well as those in whom significant intervention is considered. Since they form a minority of in-patients, their exclusion from outcome-based evaluation would not alter our conclusions regarding the use of undifferentiated NEWS2 thresholds. Nevertheless, this population warrants separate analysis, in particular those whose resuscitation status changed in the 24 hours prior to death. This may inform

an understanding of how different implementations of EWS's are associated with different in-hospital cardiac arrest rates, possibly because they prompt DNACPR decisions differently.[38]

Our analysis also excluded day 1 of elective admission. This is because different cause–effect relationships during that day significantly confound the relationship between physiological derangement and outcome. Previous studies have excluded similar cohorts by not including elective admissions,[27–29] or day-case admissions.[9 15] There was, however, no evidence that inclusion of elective admissions on day 1 following their return to the ward, would alter an overall assessment of the utility of NEWS2.

A single-centre analysis has limitations with respect to generalisation to other populations. Our underlying data are consistent with reports from other NHS acute hospitals.[27 29] Furthermore, an editorial response from 2012, employing unpublished data from users of the VitalPac EWS, suggested that RCP recommendations on NEWS (and now NEWS2) would be unsustainable across an entire in-patient population.[39] This can be understood not simply as an issue of capacity but potential jeopardy arising from the redirection of resource.[9] In an acute admissions unit, where event rates are high and resources already targeted, value may be realised from a representation of risk that supports healthcare professionals tracking of patients.[3] As already discussed, this is a different task to the efficient discrimination required of a screening test, applied across the in-patient population, to trigger further clinical evaluation. This distinction is also relevant to recommendations on the identification of sepsis, in which an NEWS2 ≥5 is used to prompt clinical assessment by a senior decision-maker.[2 40] This threshold was developed from the association between outcome and NEWS[41] or quick sequential organ failure assessment (qSOFA),[27 28] in those with features of sepsis. NEWS2 ≥5 used as a screening tool for sepsis has not been directly assessed. Our analysis illustrates the potential real-world consequences of such recommendations, including on the distribution of senior clinical resource.[9]

In summary, we identify why NEWS2 may not perform as well as expected when screening the in-patient population. This relates not just to moderate classification performance of NEWS2 in this setting (23), but the consequences of high alert rates, including competition for clinical resource[9] and clinicians' attention.[33] There is a particular need to manage multiple alerts in rapid succession.[30 35] These problems relate in part to the fact that NEWS2 was evolved for paper-based, or stand-alone implementation. In these settings, alert censoring by ward staff is well documented, whether appropriate or not.[19 20] EHR's automate alerting, thereby generating different problems, associated with high alert-rates. The EHR offers the opportunity to develop more sophisticated scoring systems incorporating a wide range of data, however, correspondence with a national, paper-based system would then be lost.[30] Any such development would require careful evaluation using a suitable methodology, such as cluster randomisation.[16 42] This approach has recently demonstrated improved 30-day mortality in patients identified to be at risk

of deterioration in Kaiser-Permanente hospitals. An algorithm was used to assess co-morbid conditions and laboratory parameters, as well as physiological parameters.[43] At the chosen response threshold, only 2.8 alerts per 100 patients per day were generated. This was followed by a complex intervention involving remote review by specially trained nurses, stipulated to avoid alert fatigue in hospital staff. It is, therefore, a significantly different implementation to NEWS2 in England, but one that demonstrates the potential for targeted assessment using an EHR. Indeed, it may be that simple scoring systems are limited in their capacity to confer net benefit across a diverse population, receiving current standards of routine review.[31] Failure to show survival benefit from EWS triggered responses would in that case, not simply reflect limitations in the methodologies used to undertake assessment.[15]

In conclusion, there is a risk that currently constituted NEWS2 based response recommendations could adversely impact the overall delivery of care to an in-patient population.[9] The response to multiple alerts requires better definition and ongoing evaluation. As a result, we would not support undifferentiated implementation of current recommendations at a key threshold NEWS2 score ≥5, across the entire in-patient population at our centre. Given existing reports of NEWS2 performance, our findings are likely to be relevant to other acute hospitals.

**Contributors** TP, GG and SB designed the study. HG, FE and SG curated the health data and conducted the analysis. ES and SB wrote the paper. TP, SG and GG contributed to manuscript revision. All authors approved the final version. SB is senior author and responsible for overall content as the guarantor.

**Funding** Health Data Research UK.

**Competing interests** TP, HG, FE and SG report no conflicts of interest. SB and GG report grant funding from HDR-UK during the conduct of the study. GG further acknowledges the support of the NIHR Birmingham ECMC, the NIHR Birmingham SRMRC and the Nanocommons H2020-EU (731032). ES reports grants from HDR-UK, during the conduct of the study; grants from Medical Research Council, grants from NIHR, grants from Wellcome Trust, grants from British Lung Foundation, grants from Alpha 1 Foundation, outside the submitted work.

**Patient consent for publication** Not applicable.

**Ethics approval** This study was approved by the East Midlands–Derby Research Ethics Committee (reference: 20/EM/0158) and Confidentiality Advisory Group (CAG: 20/CAG/0084) as part of PIONEER, the HDR-UK Hub in Acute Care.

**Provenance and peer review** Not commissioned; externally peer reviewed.

**Data availability statement** Data are available on reasonable request. The anonymised participant data and a data dictionary defining each field will be available to others through application to PIONEER, the HDR-UK Health data Hub via the corresponding author. The data will be available upon request and following approval by patients and public members (the PIONEER Data Trust Committee) and agreement of a process to ensure ethical data governance and through a data access licensing agreement. Please contact the corresponding author for details.

**ORCID iDs**
Elizabeth Sapey http://orcid.org/0000-0003-3454-5482
Felicity Evison http://orcid.org/0000-0002-9378-7548
Simon Ball http://orcid.org/0000-0001-7410-5268

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
