## [Reviewer comments · BMJ Open]

ARTICLE DETAILS

TITLE (PROVISIONAL)	An evaluation of NEWS2 response thresholds in a retrospective observational study from a UK acute hospital.
AUTHORS	Pankhurst, Tanya; Sapey, Elizabeth; Gyves, Helen; Evison, Felicity; Gallier, Suzy; Gkoutos, George; Ball, Simon

VERSION 1 – REVIEW

REVIEWER	Martín-Rodríguez, Francisco Universidad de Valladolid
REVIEW RETURNED	25-Jul-2021

GENERAL COMMENTS	I would like to thank the authors for the opportunity to review this manuscript. Overall it is well written, interesting, the subject matter is topical, and the methodology is robust. However, I have some questions that I hope the authors will answer one by one. Abstract: Line 16, describe whether it is a retrospective or prospective study Methods: clarify the main outcome, it indicates that it is the first measurement, but since when, it is not the same thing to have been admitted for 1 day and to have the set of vital signs available after 24 hours to calculate the NEWS2, as it is to calculate it on arrival at the ED. Similarly, you should indicate who makes the determinations (ERN, RN ...), where (triage box, ED, ICU) and with what devices the measurements have been made. Calibration of NEWS2: amplify and clarify these statements with concrete data. Discussion: I suggest the authors to carefully review the discussion, for non-understanding readers, it may give the impression that they are suggesting that NEWS2 is not a useful tool and wastes economic resources, when I believe that is not the idea they want to convey. Table 2. Indicate confidence intervals and likelihood ratio.
--

REVIEWER	Mackay, Jonathan Royal Papworth Hospital, c/o Anaesthesia Dept
REVIEW RETURNED	03-Sep-2021

GENERAL COMMENTS	JM Comments on BMJ Open Article submitted for peer review – Are current NEWS2 clinical response thresholds optimised for a general in-patient population? Major Strengths: I was extremely impressed by the excellent adoption and documentation of DNACPR decisions in this single centre study. Exclusion of NEWS2 recordings in patients with an active DNACPR order in place is a major potential addition to the early warning score literature. The conceivable inclusion of >100k
---

consecutive admissions to a large UK acute hospital within nine month pre-Covid timeframe is also notable.

Limitations: Low outcome rate (0.22/100 patient days). The outcome rate may have been higher if Day 1 post elective admissions had not been excluded. The authors state that 94.7% of elective admissions were followed by a planned procedure with potentially high subsequent incidences of deranged physiology - and presumably adverse events including ICU escalation.

Although not a deal breaker, I am puzzled by the decision to exclude this group of potentially vulnerable patients from outcome-based analysis. A basic premise of NEWS2 is that the score is suitable for all patients in level 1 ward areas - including post medical or surgical procedure patients. This exclusion is discussed (Page 17 Lines 4-7) 'Again, this exclusion does not imply that NEWS2 is not of use, but that a different outcome analysis would be required, arguably one beginning on return to a ward post-procedure.' Given that the authors applied this methodology to study patients discharged from ICU, I believe it would also be feasible to utilise this methodology to patients returning to the ward post procedure.

Elephant in the room with NEWS2: Is the major problem or real question

- a) With optimisation of the NEWS2 response threshold as asked in the title of this paper? – or
- b) Rather a more fundamental problem with the current NEWS2 scoring system?

I believe the fact that alert fatigue and projected clinical workload are both problematic at the lower NEWS \geq 5 threshold - with a reported sensitivity of <0.5 - points to the latter.

Limitations of NEWS2 include:

- i. relatively wide physiological dividing bins for other individual parameters with failure to recognise
 - a. increasing FiO₂ from ~0.3 to \geq 0.4
 - b. desaturation from 91% to \leq 90% or
 - c. systolic hypotension from 90 to \leq 89
- ii. suboptimal binary response to oxygen therapy
- iii. failure to distinguish between improving, stable or deteriorating physiology

Potential Strengths: Although using an alternative Standard Early Warning Score (SEWS) to standard NEWS2 to trigger Clinical Outreach is theoretically attractive to the Birmingham study design, there are clinical issues to consider. Maximum individual parameter scores of 3 for SEWS include Resp Rate \geq 36, Saturations < 85, Temperature \leq 33.9oC, BP \leq 69 systolic and pulse \leq 29. Many would consider achieving the threshold SEWS score of 4 to be a very late (rather than early) warning score.

The authors highlight the concerns of NICE regarding the clinical response recommendations for urgent clinical review at the key NEWS threshold of NEWS2 \geq 5. The authors also discuss the very high incidence of alert rates at current NEWS2 thresholds and confirm that this is also an issue in other NHS acute hospitals. International studies (including Bedoya AD et al Crit Care Med 2019) share these concerns. Bedoya et al concluded that 'an automated alert triggered by an underperforming EWS may lead to alert fatigue and breakdown of the clinical response pathway'.

	Even at the lower action NEWS2 threshold, Index-NEWS2 sensitivity was less than 0.5 suggesting that other routes to clinical evaluation, such as routine or symptom-based assessment may be necessary to identify deteriorating patients for assessment. Interestingly Zhu Y et al Resuscitation 2020 also demonstrated the Birmingham finding that higher observation frequency increased EWS performance. Zhu et al speculated that observation frequency may be a surrogate marker of increased nursing concern due to either new patient symptoms or nurse intuition. Pankhurst et al identify that the EHR offers the opportunity to develop more sophisticated scoring systems incorporating a wide range of data, but that correspondence with a national, paper-based system would then be lost. By 2022 when NEWS2 is due for review, the vast majority of acute hospitals in high income countries will have moved from 'pen and paper' to 'electronic' observation charts. This provides an opportunity for a future NEWS3 steering group to consider the case of need to upgrade from a simple additive score to a more sophisticated scoring systems – possibly including patient trajectory and observation frequency. This reviewer believes that inclusion of co-morbid conditions and laboratory parameters - as well as physiological parameters - in future sophisticated scoring systems as used in Escobar's pilot hospitals is still some way off in the UK – though it may be a potential topic for discussion by a future NEWS4 Steering group! References Bedoya AD, Clement ME, Phelan M, et al. Minimal Impact of Implemented Early Warning Score and Best Practice Alert for Patient Deterioration. Crit Care Med 2019; 47: 49–55 Zhu Y, Chiu YD, Villar SS, et al. Dynamic individual vital sign trajectory early warning score (DyniEWS) versus snapshot national early warning score (NEWS) for predicting postoperative deterioration. Resuscitation 2020; 157: 176-184
--	--

REVIEWER	Haegdorens, Filip University of Antwerp, Nursing and Midwifery Sciences
REVIEW RETURNED	14-Sep-2021

GENERAL COMMENTS	I would like to thank the editor and authors for providing this very relevant paper concerning NEWS2 clinical response thresholds. Studying the actual performance of NEWS in clinical practice is crucial since these systems seem to produce a significant amount of false positive alerts leading to alarm-fatigue. I want to congratulate the authors on the way data was handled since they ensured that an outcome event was associated with the preceding NEWS2 score within a 24-hour timeframe. From own experience, I know that handling and analysing time dependent data can be challenging. The method section is very clear and concise. The results are clearly presented and contain detailed and relevant analyses. I think the authors succeeded in their attempt to analyse the theoretical impact of using different thresholds for NEWS2 on alert rate / NNE and the prediction of events. The discussion is generally well written and provides insight in the results. I agree with the authors' statement on the risks of implementing a key threshold of ≥ 5 across the entire hospital. This manuscript is ready for publication.
--

	Comments - The authors found a low event rate in the group of patients that were included in the outcome analysis because they excluded DNACPR patients. They mentioned that previous reports have excluded patients on end-of-life care pathways and that this had to be inferred from an absence of observations in the 24 hours prior to death. However, in our study from 2018 (https://doi.org/10.1016/j.resuscitation.2018.04.018) we also excluded all DNACPR patients and patients receiving end-of-life care by reviewing patient records manually. Could the authors provide some more information on the DNACPR policy in their hospital and the adherence to this policy? Is it possible that non-DNACPR patients could have received and-of-life care? - How relevant is the outcome 'ICU admission'? Were there any patients who were admitted to the ICU for other reasons than deterioration? In some hospitals, patients are admitted to the ICU after for example a diagnostic procedure. Were these non-critical transfers to the ICU included in the outcome variable? - We also calculated, albeit using the maximum NEWS in the 24 hours before an event, the performance of the NEWS and found a PPV of 6,76% and NPV of 99,52% for NEWS ≥ 5 (https://doi.org/10.1111/jocn.15493). However, the PPV in this study was much lower (1.90%). How do the authors explain this difference? Could it be due to the difference in study design and sample, NEWS vs NEWS2, or due to the selection of outcome variables? We found that 12-13% of NEWS2 scores were positive (≥ 5) which corresponds with the numbers in this study (p12 line 58: 12.3 /100 patients /day). - p9 line55: Holland is a geographical region in the Netherlands. I would change this to 'The Netherlands'
--	---

VERSION 1 – AUTHOR RESPONSE

Reviewer: 1

Dr. Francisco Martín-Rodríguez, Universidad de Valladolid

Comments to the Author:

I would like to thank the authors for the opportunity to review this manuscript. Overall, it is well written, interesting, the subject matter is topical, and the methodology is robust.

Response: We thank the reviewer for these kind words.

However, I have some questions that I hope the authors will answer one by one.

Abstract: Line 16, describe whether it is a retrospective or prospective study

Response: We thank the reviewer for this point which has now been addressed by explicitly specifying the study design as follows:

'Design: Retrospective Observational Cohort Study'

Reviewer: Methods: clarify the main outcome, it indicates that it is the first measurement, but since when, it is not the same thing to have been admitted for 1 day and to have the set of vital signs available after 24 hours to calculate the NEWS2, as it is to calculate it on arrival at the ED.

Response: We thank Dr. Martín-Rodríguez for this comment.

The main outcome was death or admission to intensive care unit within 24 hours of a NEWS2 score. The NEWS2 score included in the assessment of outcome was termed the 'Index' NEWS2 (first taken within a 24 hour period). NEWS2 is calculated automatically by the electronic health records, and so are available as soon as physiological observations are taken. NEWS2 recordings are mandated in UK hospitals with KPIs to ensure recordings are measured early during a patient's first contact with the hospital. If the patient was admitted during the evening or night, the 24-hour period would reflect this starting time. We also wish to clarify that, following this first 24-hour period, all subsequent Index NEWS2 were included, being also the first NEWS2 reading taken in subsequent 24-hour time periods.

In response, we have added a sentence to the following paragraph (***revisions shown in bold italic throughout***) in order to clarify that all days other than Day 1 post elective admission are included:

'Day 1 post elective-admission was excluded from outcome-based analysis because it was known that almost all were admitted for surgery later that day, after a NEWS2 was recorded. Any relationship with outcome might then be confounded by decisions not to proceed to intervention on that day, informed by NEWS2 or its component observations. ***All other patient days were eligible for analysis.***'

We agree that the performance of NEWS2 may vary according to route of and time post admission. On Day 1 post-elective admission the relationship between score and outcome is confounded by the occurrence of elective procedures (surgery), hence our decision to exclude this from the analysis of NEWS2 performance. This is discussed in greater detail in our response to Reviewer 2. We have included all other patient-days since the aim was to closely represent the real-world implications of a hospital wide implementation of NEWS2 and associated response recommendations.

Although we observed some differences in performance according to route and time of admission (e.g. c-statistic for elective admissions after Day 1 = 0.70 and for Day 1 of emergency admission =

0.81) and outcome event rate (0.40% per patient-day on the first day post-emergency admission compared to 0.22% per patient-day overall), given the distribution of patients shown in Table S4 (98 ± 15 patients occupying a bed were emergency admissions in hospital for < 24 hours, 632 ± 31 emergency admissions in hospital for ≥ 24 hours, 198 were elective admission in hospital ≥24 hours), even if the case mix were adjusted to be entirely emergency admissions, the impact on our overall assessment of NEWS2 performance would be small.

Reviewer: Similarly, you should indicate who makes the determinations (ERN, RN ...), where (triage box, ED, ICU) and with what devices the measurements have been made.

Response: We thank Dr. Martín-Rodríguez for this comment, in response to which we have now added a sentence to the Methods section which reads as follows:

'NEWS2 observations were recorded by trained healthcare staff with devices used and maintained in accordance with Medicines Health Regulation Agency guidance [21].'

Reference 21 has been added to the relevant MHRA guidance.

The location of these procedures is described by the following sentences in the Methods section:

'Initial Emergency Department assessments, prior to admission, were not included.'

'NEWS2 scores were not eligible for inclusion if at the time the score was acquired, the patient was in ICU or had a DNACPR in place'.

Following the decision to admit through the Emergency Department the first set of observations were analysed. This could occur whilst in the Emergency Department or following transfer to an in-patient ward.

Devices include a number of different blood pressure machines and oximeters. Of note, all equipment has been validated as medical equipment for use in NHS care providers and a note to this effect has been made in the publication methods section.

Reviewer: Calibration of NEWS2: amplify and clarify these statements with concrete data.

Response: We thank Dr. Martín-Rodríguez for pointing this out and we apologise for the lack of clarity. As commented upon by others (for example in Reference 27), NEWS2 does not estimate absolute risk of an outcome event, so that it is not strictly possible to undertake a calibration exercise.

Statistical modelling, such as that reported in Reference 42 estimates absolute risk. Indeed, as Dr. Martín-Rodríguez mentions above, the risk associated with any given NEWS2 is likely to be context dependent. As Reviewer 2 discusses below, that context will be increasingly accounted for as detailed representations of the clinical phenotype are captured in any data model. It will then become increasingly possible to estimate absolute risk in the ways demonstrated in Reference 42, rather than on physiological observations alone.

(This is not to say that an internal calibration exercise does not have value when developing a scoring system, but this was not the purpose of our paper).

We have, in a different context, been asked to represent the incremental change in risk associated with progressive changes in EWS action thresholds as in Fig S3. The progressive stepwise representation of risk by NEWS2 is one of its more important attributes and we believe this illustration, using data from a diverse in-patient population, may be of interest to the reader. We realise that we have not worded the paragraph sufficiently well. In response to the reviewer's comment, we have revised the following paragraph to reflect the points raised:

Calibration was not assessed as NEWS2 does not estimate absolute risk [27]. Usefully, integer changes in NEWS2 thresholds were associated with approximately equal relative changes in the odds ratio of outcome events, for a wide range of scores (Fig S3).

We have also expanded Fig S3a legend a little as follows with changes in bold italic:

The Ln of the Odds Ratio (Ln OR) \pm 95% confidence intervals for occurrence of the composite outcome event when the Index NEWS2 \geq threshold score vs. $<$ threshold score. The composite outcome event was the first of unplanned admission to ICU (type 1 and 2 of the NHS critical care minimum dataset) or death of the patient within 24 hours of a NEWS2 score. This analysis included NEWS2 scores from Day 1 to 56 post-admission other than for Day 1 of elective admission (***Table S2a and S2b***).

Although NEWS2 is not constituted to report absolute risk, it exhibits features ***useful*** in representing risk. Integer changes in threshold were associated with ***an approximately equal change in the odds ratio of an outcome event. Across the range of scores, the OR of an event \geq threshold score vs. $<$ threshold score, increased by a factor of 1.2. (Ln OR of composite outcome event = 1.35 + 0.205 (Index NEWS2 threshold score); $R^2 = 0.99$. For increase in NEWS2 = 1, the increase in Ln OR of composite outcome event = 0.205, the OR of composite outcome event increases by a factor of $e^{0.205} = 1.2$).***

(These relationships resemble Fig 2b, because the relationship between sensitivity and Index NEWS2 is approximately negative linear and because at low event rates the odds of an event at threshold \approx PPV = 1/NNE; the Ln (odds of an event) \approx -Ln (NNE)).

We have also added a row to each of Table S2a, S2b, S3a and S3b in order to make more apparent the basis for calculation of the odds ratios shown in Fig S3a and S3b. These show the cumulative total at or below the NEWS2 score with or without an event.

However, if Dr. Martín-Rodríguez feels that this remains problematic, we would be happy for this paragraph to be removed from the manuscript since as commented in the Fig S3 legend, this is simply a different way of representing the data already shown.

Reviewer: Discussion: I suggest the authors to carefully review the discussion, for non-understanding readers, it may give the impression that they are suggesting that NEWS2 is not a useful tool and wastes economic resources, when I believe that is not the idea they want to convey.

Response: We thank Dr. Martín-Rodríguez for this suggestion. Our manuscript does not set out to suggest that NEWS2 does not have value, as a progressive representation of risk across any given in-patient population. Our point is that literal interpretation of current UK recommendations reflects over optimistic assumptions regarding the application of NEWS2 in the real world, which we have attempted to evaluate and discuss. Indeed, Reviewer 2 considers that there may be a more fundamental problem with the current NEWS2 scoring system, within the context of its use as a screening tool. It is for this reason that we carefully based our discussion on the NICE exceptional surveillance report from 2019. This includes acknowledgement of the potential for unexpected adverse consequences arising from response recommendations at current thresholds, resulting from competition for human resource.

We are of the opinion that NEWS2 is currently the best available early warning score, albeit marginally. However, it is within the bounds of possibility that, literally interpreted, the current RCP / NHS recommendations could be counterproductive. This seems to be accepted in the aforementioned NICE exceptional surveillance report and is acknowledged by Reviewer 2 and 3 in their reviews, and publications.

In our discussion, we have attempted to point out that the issue sits with the use case defined by current response recommendations. Arguably the scoring system is as good as can be achieved with this type of methodology (as detailed by Reviewer 2). In response to the concerns of Reviewer 1 we have identified the following sentence that might be interpreted differently, so have revised the precise wording.

Diversion of resource is one mechanism by which NEWS2 implementation could adversely affect patient care, particularly if alert rates are high [9].

is changed to

Diversion of resource is one mechanism by which NEWS2 associated response recommendations could adversely affect patient care, particularly if alert rates are high [9].

Table 2. Indicate confidence intervals and likelihood ratio.

Response: We thank Dr. Martín-Rodríguez for pointing this out to use and, in response, we have added Likelihood Ratio and Confidence Intervals in our revised manuscript.

Reviewer: 2

Dr. Jonathan Mackay, Royal Papworth Hospital

Comments to the Author:

JM Comments on BMJ Open Article submitted for peer review – Are current NEWS2 clinical response thresholds optimised for a general in-patient population?

Major Strengths:

Reviewer: I was extremely impressed by the excellent adoption and documentation of DNACPR decisions in this single centre study. Exclusion of NEWS2 recordings in patients with an active DNACPR order in place is a major potential addition to the early warning score literature. The conceivable inclusion of >100k consecutive admissions to a large UK acute hospital within nine month pre-Covid timeframe is also notable.

Response: We thank Dr. Mackay for this appreciation.

Limitations:

Reviewer: Low outcome rate (0.22/100 patient days). The outcome rate may have been higher if Day 1 post elective admissions had not been excluded. The authors state that 94.7% of elective admissions were followed by a planned procedure with potentially high subsequent incidences of deranged physiology - and presumably adverse events including ICU escalation.

Reviewer: Although not a deal breaker, I am puzzled by the decision to exclude this group of potentially vulnerable patients from outcome-based analysis. A basic premise of NEWS2 is that the score is suitable for all patients in level 1 ward areas - including post medical or surgical procedure patients.

Response: We thank Dr. Mackay for this comment. As he suggests the event rate in patients who are for resuscitation is low at 0.22/100 patient days. As discussed in our response to Reviewer 3, overall event rates are however similar to those observed in their multicentre study in Belgium reported in 2018.

We appreciate Dr. Mackay's asking the question regarding Day 1 post-elective admission. Indeed, it was discussed in more detail in an earlier version of our manuscript, but this was subsequently omitted so as to present a simple analytical narrative. We examined Day 1 post elective admissions in a sensitivity analysis. If early post-procedure data are analysed, however selected, our conclusions regarding the limitations of NEWS2 remain valid. In fact the limitations of NEWS2 based screening are if anything accentuated. In particular:

- **In the Day 1 post elective admission population, the event rate was 0.37 per 100 patient days.**

This reflects a heterogeneous population, from minor surgery with a length of stay less than 1 day through to major elective surgery with planned ICU admission (Type 4 critical care admission code).

As discussed in our response to Reviewer 3, planned admissions to ICU would not generate an outcome event.

- **The c-statistic for Index-NEWS2 on Day 1 of elective admission = 0.51.**

This was as anticipated, since the vast majority of elective patients are intentionally admitted with little measurable physiological derangement. Indeed, there is a chance for bias to arise from decisions not to proceed to planned intervention on the basis of pre-operative observations. This was the basis for our original plan of analysis. (It therefore resembles your Zhu paper in which pre-operative observations were not evaluated).

- **In post-hoc analysis, we found that if the first NEWS2 > 12 hours post elective admission is used the c-statistic = 0.63 and at > 24 hours post elective admission = 0.60.**

After the first 24 hours, NEWS2 performance improves such that the c-statistic for elective admissions trends towards those for emergency admissions, resulting in the c-statistic = 0.70 for Index NEWS2 in elective admissions excluding Day 1 or c-statistic = 0.74 for Index NEWS2 in elective admissions excluding Day 1 and 2.

As Dr Mackay anticipates the event rate associated with the first NEWS2 > 12 hours post elective admission was indeed higher at 0.71/100 patient days, however the event rate then falls, so that associated with first NEWS2 > 24 hours post elective admission was 0.34 /100 patient days.

(It should also be noted that the proportion of beds occupied by this group of patients is relatively low so that the impact on overall assessment of performance is to an extent limited).

In summary, the performance of NEWS2 throughout the day of elective admission was poor and the event rate perhaps not as high as the Dr. Mackay might have expected. This will of course be influenced by case mix, for example this population is different to the analysis of cardiac surgical wards in the Zhu paper. Although beyond the scope of this paper, our data suggests that the discriminatory power of a 'snapshot' NEWS2 in the hours following a procedure might not be as high as expected, perhaps because the situation is so highly dynamic. It is presumably for this reason that Dr Mackay and colleagues worked on DyniEWS. It is perhaps not entirely surprising that early post elective admission NEWS2 performs less well than following acute admission, given NEWS2 origins in NEWS and ViEWS which we understand were initially developed and validated on datasets from acute admission pathways.

To address Dr. Mackay's comment, we have added further supplementary data linked to the results section of our revised manuscript. This additional material is now Fig S2a and the previous Fig S2 becomes Fig S2b.

The following is added to the results:

(The performance of NEWS2 across day 1 post elective admission is shown in Fig S2a)

The following is added to the online supplementary figures:

Online Supplementary Figure 2a (Fig S2a)

Legend Fig S2a

Receiver operating characteristic (ROC) for the first NEWS2 obtained from 0-, 12- and 24-hours post elective-admission. The respective c-statistics were 0.51, 0.63 and 0.60. The c-statistic = 0.74 for later post elective admission patient-days combined. The outcome event was the first of unplanned admission to ICU (type 1 and 2 of the NHS critical care minimum dataset) or death of the patient within 24 hours of an evaluated NEWS2 score. As anticipated in the study design, Index NEWS2 on Day 1 of elective admission was not predictive of an adverse outcome. Its acquisition almost always preceded physiological insult, specifically a planned procedure occurred in 94.7% of patients in the subsequent 12 hours. A detailed assessment of the discriminatory power of NEWS2 in the hours immediately following a procedure is beyond the scope of this analysis but appears low.

The NEWS2 associated event rate on Day 1 post elective-admission = 0.37/100 patient-days and Day 2 = 0.34/100 patient-days.

Reviewer: This exclusion is discussed (Page 17 Lines 4-7) ‘Again, this exclusion does not imply that NEWS2 is not of use, but that a different outcome analysis would be required, arguably one beginning on return to a ward post-procedure.’ Given that the authors applied this methodology to study patients discharged from ICU, I believe it would also be feasible to utilise this methodology to patients returning to the ward post procedure.

Response: We apologise for confusion caused by our efforts to avoid drawing conclusions from data not included in the original analysis. (Given some knowledge of performance during Day 1 post elective admission). Our rationale was that there might be a sub-group in whom the event rate may be higher and for which NEWS2 performance was better, such as those admitted to a post-anaesthetic care unit, but which we were not powered to address.

As is now shown above, inclusion of NEWS2 from all Day 1 elective admissions, at an alternative time point that would capture the early post-operative period, does not alter our conclusions. Given the limitations that Dr Mackay’s group identified when evaluating NEWS post cardiac surgery, our finding in a diverse population will probably be no surprise. As he will appreciate, considering Day 1 post-elective admission in any more detail becomes an entirely separate analysis, in which the potential to introduce other selection bias needs to be considered. (For example, one might expect criteria for transfer from post-op recovery back to the ward to influence the NEWS2 set obtained immediately on return).

We have revised the following paragraph in our manuscript:

'Our analysis also excluded Day 1 of elective admission. This is because different cause-effect relationships during that day significantly confound the relationship between physiological derangement and outcome. Previous studies have excluded similar cohorts by not including elective admissions [24-26], or day-case admissions [9, 15]. Again, this exclusion does not imply that NEWS2 is not of use, but that a different outcome analysis would be required, arguably one beginning on return to a ward post-procedure.'

To the following

*'Our analysis also excluded Day 1 of elective admission. This is because different cause-effect relationships during that day significantly confound the relationship between physiological derangement and outcome. Previous studies have excluded similar cohorts by not including elective admissions [24-26], or day-case admissions [9, 15]. **There was however no evidence that inclusion of elective admissions on Day 1, following their return to the ward, would alter an overall assessment of the utility of NEWS2.***

Reviewer: Elephant in the room with NEWS2: Is the major problem or real question

a) With optimisation of the NEWS2 response threshold as asked in the title of this paper? – or

b) Rather a more fundamental problem with the current NEWS2 scoring system?

I believe the fact that alert fatigue and projected clinical workload are both problematic at the lower NEWS₂≥5 threshold - with a reported sensitivity of < 0.5 - points to the latter.

Limitations of NEWS2 include:

i. relatively wide physiological dividing bins for other individual parameters with failure to recognise

a. increasing FiO₂ from ~0.3 to ≥0.4

b. desaturation from 91% to ≤90% or

c. systolic hypotension from 90 to ≤89

ii. suboptimal binary response to oxygen therapy

iii. failure to distinguish between improving, stable or deteriorating physiology

Potential Strengths: Although using an alternative Standard Early Warning Score (SEWS) to standard NEWS2 to trigger Clinical Outreach is theoretically attractive to the Birmingham study design, there are clinical issues to consider. Maximum individual parameter scores of 3 for SEWS include Resp Rate ≥ 36, Saturations < 85, Temperature ≤ 33.9oC, BP ≤ 69 systolic and pulse ≤ 29. Many would consider achieving the threshold SEWS score of 4 to be a very late (rather than early) warning score.

Response: We thank Dr. Mackay for raising these points, with which we agree. We admit to being cautious in choosing to frame our question as a) rather than b). It is interesting that Dr Mackay has identified b) with which we clearly have some sympathy, as commented upon in the penultimate paragraph of our discussion.

As for the SEWS alert question, Dr Mackay will be aware that the SEWS alert at 4 reflected some previous recommendations. In comparing NEWS2 and SEWS on this dataset we found that there was a small difference in performance in favour of NEWS2.

Irrespective of the pros and cons of SEWS, the considerably lower rate of alerting at the SEWS threshold reduces the likelihood of confounding outcome analysis by intervention. As Dr. Mackay alludes to, we think this is a strength of the study. The purpose of our evaluation was to determine whether there was any justification in effectively lowering the threshold of alerting.

Our own experience, including now with NEWS2, is that there are issues relating to the failure of clinicians to understand the limitations of using an EWS as a negative predictor, in the face of clinically important abnormalities in single parameters.

Reviewer: The authors highlight the concerns of NICE regarding the clinical response recommendations for urgent clinical review at the key NEWS threshold of NEWS2 \geq 5. The authors also discuss the very high incidence of alert rates at current NEWS2 thresholds and confirm that this is also an issue in other NHS acute hospitals. International studies (including Bedoya AD et al Crit Care Med 2019) share these concerns. Bedoya et al concluded that ‘an automated alert triggered by an underperforming EWS may lead to alert fatigue and breakdown of the clinical response pathway’.

Response: We thank Dr. Mackay for suggesting this reference which we have now included in our revised manuscript.

Reviewer: Even at the lower action NEWS2 threshold, Index-NEWS2 sensitivity was less than 0.5 suggesting that other routes to clinical evaluation, such as routine or symptom-based assessment may be necessary to identify deteriorating patients for assessment. Interestingly Zhu Y et al Resuscitation 2020 also demonstrated the Birmingham finding that higher observation frequency increased EWS performance. Zhu et al speculated that observation frequency may be a surrogate marker of increased nursing concern due to either new patient symptoms or nurse intuition.

Response: We thank Dr. Mackay for this point. We agree with the paper's speculation on the significance of frequency of measurement. Our collective findings go beyond the conclusion that frequency of monitoring is

an additional independent predictor of risk (or event rate), to the fact that NEWS2 performs differently in this group of patients. (Otherwise, dimensionless metrics like sensitivity and specificity would not vary with the frequency of measurement). This is slightly different from the frequency of observation

being independently associated with outcome, albeit a related point. We think this is important when interpreting earlier literature on early warning score performance using all sets of observations.

We have added the Zhu reference.

Reviewer: Pankhurst et al identify that the EHR offers the opportunity to develop more sophisticated scoring systems incorporating a wide range of data, but that correspondence with a national, paper-based system would then be lost. By 2022 when NEWS2 is due for review, the vast majority of acute hospitals in high income countries will have moved from ‘pen and paper’ to ‘electronic’ observation charts. This provides an opportunity for a future NEWS3 steering group to consider the case of need to upgrade from a simple additive score to a more sophisticated scoring systems – possibly including patient trajectory and observation frequency. This reviewer believes that inclusion of co-morbid conditions and laboratory parameters - as well as physiological parameters - in future sophisticated scoring systems as used in Escobar’s pilot hospitals is still some way off in the UK – though it may be a potential topic for discussion by a future NEWS4 Steering group!

Response: We thank Dr. Mackay for his comment with which we agree. It might not be surprising that we are also working on more extensive analyses. We have read with interest the DyniEWS paper. (Our original manuscript was prepared before this was published).

References

Bedoya AD, Clement ME, Phelan M, et al. Minimal Impact of Implemented Early Warning Score and Best Practice Alert for Patient Deterioration. Crit Care Med 2019; 47: 49–55

Zhu Y, Chiu YD, Villar SS, et al. Dynamic individual vital sign trajectory early warning score (DyniEWS) versus snapshot national early warning score (NEWS) for predicting postoperative deterioration. Resuscitation 2020; 157: 176-184

Reviewer: 3

Dr. Filip Haegdorens, University of Antwerp

Comments to the Author:

I would like to thank the editor and authors for providing this very relevant paper concerning NEWS2 clinical response thresholds. Studying the actual performance of NEWS in clinical practice is crucial since these systems seem to produce a significant amount of false positive alerts leading to alarm-fatigue. I want to congratulate the authors on the way data was handled since they ensured that an outcome event was associated with the preceding NEWS2 score within a 24-hour timeframe. From own experience, I know that handling and analysing time dependent data can be challenging. The method section is very clear and concise. The results are clearly presented and contain detailed and relevant analyses. I think the authors succeeded in their attempt to analyse the theoretical impact of using different thresholds for NEWS2 on alert rate / NNE and the prediction of events. The discussion is generally well written and provides insight in the results. I agree with the authors' statement on the risks of implementing a key threshold of ≥ 5 across the entire hospital. This manuscript is ready for publication.

Response: We thank Dr. Haegdorens for his kind comments.

Comments

Reviewer: The authors found a low event rate in the group of patients that were included in the outcome analysis because they excluded DNACPR patients. They mentioned that previous reports have excluded patients on end-of-life care pathways and that this had to be inferred from an absence of observations in the 24 hours prior to death. However, in our study from 2018 (<https://doi.org/10.1016/j.resuscitation.2018.04.018>) we also excluded all DNACPR patients and patients receiving end-of-life care by reviewing patient records manually.

Response: We thank Dr. Haegdorens for this comment and for bringing this study to our attention. It is an important finding which we have now included in the references our revised manuscript.

Reviewer: Could the authors provide some more information on the DNACPR policy in their hospital and the adherence to this policy? Is it possible that non-DNACPR patients could have received end-of-life care?

Response: We thank Dr. Haegdorens for raising these points. Our electronic healthcare records represent each patient's resuscitation status and any treatment limitation and escalation (TEAL) decisions. This is shown continuously in the individual patient banner. Furthermore, our clinical decision support system ensures that each patient's DNACPR status is regularly reviewed.

Rather than policy, this might be seen as a matter of custom and practice at our centre, in which such computable records of decision making are integral to the clinical record. A failure to then deliver appropriate care through the record would be addressed as a matter for clinical governance and quality management, not as a failure of the record, but as a failure of appropriate care delivery.

Although difficult to prove a negative assumption, the low proportion of death on the ward without a DNACPR in place, suggests that there is high fidelity correspondence between the DNACPR record and clinical decision making. It therefore seems unlikely that there are patients receiving end of life care who do not have a DNACPR in place.

Specifically, we note that in the control group from your 2018 paper, the ward mortality rate was 12.5/1000 admissions, the rate without a DNACPR code was 7.3/1000 admissions but following review of the record, the rate of unexpected death was 1.5/1000 admissions. This compares with a ward mortality of 10.7/1000 admissions and death without DNACPR code of 1.3/1000 admissions in our study.

The concordance in mortality suggests that these two populations are similar and that between DNACPR in our study and unexpected death in your study suggests that these are likely to be operationally equivalent.

In response to this comment, we have added a clause in the Methods specifying the presence of clinical decision support systems to ensure regular update of the resuscitation status:

'The EHR at QEHB (PICS, Birmingham Systems) contains time-stamped, structured records that include demography, location, time of admission and discharge, physiological measurements supporting NEWS2 and Standard Early Warning Score (SEWS) (Table S1) and Do Not Attempt Cardiopulmonary Resuscitation (DNACPR) decisions, ***subject to regular review underpinned by the clinical decision support system***'.

Reviewer: How relevant is the outcome 'ICU admission'? Were there any patients who were admitted to the ICU for other reasons than deterioration? In some hospitals, patients are admitted to the ICU after for example a diagnostic procedure. Were these non-critical transfers to the ICU included in the outcome variable?

Response: We used administrative data defined by trained coders at the time of treatment, against an NHS standard to define 'unplanned admission to ICU (type 1 and 2 of the NHS critical care minimum dataset)' in the methods section. In response to this point, we have now included a reference linking to the NHS data model and dictionary in our manuscript.

https://datadictionary.nhs.uk/attributes/critical_care_admission_type.html

For ease of reference, we show the relevant table below:

0	Unplanned local admission. All emergency or urgent PATIENTS referred to the unit only as a result
1	of an unexpected acute illness occurring within the local area (hospitals within the Trust together with neighbouring community units and services).

0 2	Unplanned transfer in. All emergency or urgent PATIENTS referred to the unit as a result of an unexpected acute illness occurring outside the local area (including private and overseas Health Care Providers).
0 3	Planned transfer in (tertiary referral). A pre-arranged admission to the unit after treatment or initial stabilisation at another Health Care Provider (including private and overseas Health Care Providers) but requiring specialist or higher-level care that cannot be provided at the source hospital or unit.
0 4	Planned local surgical admission. A pre-arranged surgical admission from the local area to the to the unit, acceptance by the unit must have occurred prior to the start of the surgical procedure and the procedure will usually have been of an elective or scheduled nature. For example, following a major procedure, for a high risk medical condition associated with any level of surgery, admitted prior to elective surgery for optimisation, admitted for monitoring of pain control eg epidurals, or obstetric surgical cases admitted on a planned basis.
0 5	Planned local medical admission from the local area. Booked medical admission, for example, planned investigation or high risk medical treatment.
0 6	Repatriation. The PATIENT is normally resident in your local area and is being admitted or readmitted to your unit from another hospital (including overseas Health Care Providers). This situation will normally arise when a PATIENT is returning from tertiary or specialist care.

Our definition would not include patients who were planned for transfer after a procedure. These would be coded as type 4 or 5.

Reviewer: We also calculated, albeit using the maximum NEWS in the 24 hours before an event, the performance of the NEWS and found a PPV of 6,76% and NPV of 99,52% for NEWS ≥ 5 (<https://doi.org/10.1111/jocn.15493>). However, the PPV in this study was much lower (1.90%). How do the authors explain this difference? Could it be due to the difference in study design and sample, NEWS vs NEWS2, or due to the selection of outcome variables?

We found that 12-13% of NEWS2 scores were positive (≥ 5) which corresponds with the numbers in this study (p12 line 58: 12.3 /100 patients /day).

Response: We thank Dr, Haegdorens for this raising this point. For the highest NEWS2 in each 24-hour period we calculate the following for our study and his study

	Our study	Haegdorens
Sensitivity	0.702	0.659
Specificity	0.879	0.887
Event Rate (Pre-test probability)	0.30%	1.23%

The difference in PPV is therefore almost entirely attributable to the difference in event rate. Indeed, the consistency of sensitivity and specificity in the different populations is reassuring, despite the

difference in methodology. (We understand your 2020 paper used the case control methodology based on Mercaldo and colleagues)

Although the maximum NEWS2 in the 24 hours prior to an SAE are analysed in your 2020 paper, we think that the event rate used, as in your 2018 paper, is a rate denominated on an admission, not a 24-hour period. Given the mean length of stay in our population this would make the event rates highly comparable.

Reviewer: p9 line55: Holland is a geographical region in the Netherlands. I would change this to 'The Netherlands'

Response: We thank Dr, Haegdorens and apologise for this error which was a source of debate amongst the authors. We stand corrected by our Reviewer from Flanders!

Reviewer: 1

Competing interests of Reviewer: No

Reviewer: 2

Competing interests of Reviewer: Researcher with an interest in early warning scores

Reviewer: 3

Competing interests of Reviewer: No competing interests

VERSION 2 – REVIEW

REVIEWER	Mackay, Jonathan Royal Papworth Hospital, c/o Anaesthesia Dept
REVIEW RETURNED	25-Nov-2021

GENERAL COMMENTS	Thank you to the authors for their detailed and helpful responses to my previous comments. I confirm that I am very happy with the revised title and editorial revisions. I am particularly grateful for the comprehensive added further supplementary data regarding the performance of NEWS2 across day 1 post elective admission in Fig S2a. I suspect many members of acute NHS 'hospital at night teams' will share the expressed sentiment in response to Reviewer 1 that 'Literal interpretation of current UK recommendations reflects over optimistic assumptions regarding the application of NEWS2 in the real world ...' Exclusion of NEWS2 recordings in patients with an active DNACPR order in place is a major strength of this study. I look forward to reading further important follow up papers from the Birmingham group.
---